# Drug Therapy Safety in Palliative Care—Pharmaceutical Analysis of Medication Processes in Palliative Care

**DOI:** 10.3390/pharmacy11050160

**Published:** 2023-10-07

**Authors:** Lisa Krumm, Claudia Bausewein, Constanze Rémi

**Affiliations:** 1Department of Palliative Medicine, University Hospital, Ludwig-Maximilians-University, 81377 Munich, Germany; 2Helios Dr. Horst Schmidt Hospital, 65199 Wiesbaden, Germany

**Keywords:** clinical pharmacists, drug-related side effects and adverse reactions, drug therapy, palliative treatment, pharmacists

## Abstract

Pharmacotherapy plays a crucial role in symptom management in palliative care and is associated with risks potentially leading to drug-related problems (DRP). Pharmacists can identify DRPs and advise prescribers on optimizing drug therapy. The aim of this study was to identify DRP in a palliative care unit (PCU) and evaluate corresponding pharmaceutical interventions. A non-randomized before-and-after study in a PCU starts with a control phase, an interphase, and an intervention phase. Primary endpoint: DRP, including pharmaceutical interventions and their acceptance. The medication of all inpatients was recorded at set time points, assessed for potential and manifest DRP, and categorized. In the control phase, the ward pharmacist did not interfere with the clinical team. In the intervention phase, the pharmacist could intervene when a DRP was identified and give recommendations. During the 12-month period, 284 patients were included (control phase n = 138; intervention phase n = 146) and 1079 DRPs were identified (control phase n = 634; intervention phase n = 445). The number of DRPs/patient was significantly reduced by the pharmacist’s interventions between the control and intervention phases (4 vs. 3 DRPs, *p* = 0.001). Overall acceptance of pharmaceutical interventions by prescribers was very high (227/256; 88%). DRPs are hardly preventable. With a clinical pharmacist as a member of the palliative care team, it is possible to reduce the number of DRPs and identify potential problems earlier.

## 1. Introduction

Palliative care generally involves a holistic approach, addressing physical, psychological, social, and spiritual needs to improve the quality of life of patients with advanced illness [1]. Pharmacotherapy plays a fundamental role in palliative care. It focuses on providing relief from symptoms and improving the overall quality of life for patients with advanced, life-limiting illnesses [1]. The goal is to manage symptoms such as pain, nausea, breathlessness, anxiety, and depression. Regular assessments and adjustments to drug therapy are crucial to ensuring ongoing symptom control and patient comfort [2]. Particularly in the last days to weeks of life, the dynamics of the disease progression can be challenging—overall and also in regard to the pharmacotherapy. Besides the desired positive influence on distressing symptoms, there are also risks associated with drug therapy. Medications for symptom control, but also comorbidities, can lead to polypharmacy (>5 drugs) in palliative care patients, which further increases the risk of interactions or side effects [3,4]. The latter play an important role in palliative care and can have an impact on patient morbidity and mortality. Distinguishing between adverse effects of prescribed medications and symptoms of progressive disease is, however, difficult and probably often ambiguous [5,6,7].

Many national and international studies have already demonstrated that the presence and interventions of a pharmacist on the ward have many positive effects, including the detection and reduction of drug-related problems (DRP) at an early stage [8,9,10]. Pharmacists have the greatest impact by directly participating in ward rounds and providing advice and direct intervention. Under- or overdoses or missed doses could be avoided through the presence of pharmacists [11,12,13]. Although pharmacists are known to reduce drug-related problems (DRPs) and potentially inappropriate medication (PIMs), data on the impact of hospital pharmacists in palliative care in general and in Germany in particular is scarce.

Therefore, the aim of this study was to identify DRP in a palliative care unit (PCU) and evaluate corresponding pharmaceutical interventions.

## 2. Materials and Methods

### 2.1. Design

Non-randomised observational before-and-after study, starting with a control phase, followed by an interphase, and concluding the intervention phase (Figure 1) [14]. This study is reported based on the STROBE checklist [14].

In the control phase, the “before” situation was presented. A pharmacist was integrated into the team of a palliative care unit and observed the medication process on the unit. The pharmacist was not allowed to intervene in this phase.

The interphase involved analyzing data from the control phase and identifying weaknesses in the medication process. Strategies to improve the process were developed and discussed with the team.

In the subsequent intervention phase (“after” situation), the pharmacist again observed the medication process but was allowed to actively intervene and make recommendations. The interphase served to evaluate the data from the control phase and develop strategies for optimization.

### 2.2. Setting

The study was conducted on the 11-bedded palliative care unit in a generalist hospital in Germany. Annually, about 300 patients with far-advanced life-limiting conditions are admitted to the unit from the community, nursing homes, and other hospital wards for the management of distressing symptoms. The multi-professional team consists of physicians, nurses, physiotherapists, psychologists, chaplains, and social workers. The average length of stay in 2018 was 9.5 days. After stabilization and symptom control, patients are discharged to the community, hospice, or other institutions, or die on the ward.

At the time of the study, the hospital was supplied with drugs by a pharmacy. No pharmaceutical services in the form of anamnesis, interaction checks, or similar were provided. Prior to our study, clinical pharmacists were not part of the team. This was due to a lack of funding.

### 2.3. Population

#### 2.3.1. Inclusion Criteria

all patients with incurable malignant or non-malignant disease admitted to the palliative care unit.Minimum age: 18 years.

#### 2.3.2. Exclusion Criteria

Patient or carer not fluent in German or English.

### 2.4. Patient Information

The treating physician informed all patients who had been admitted to the PCU and who fulfilled the inclusion criteria about the study during their visit. The pharmacist introduced herself personally to the patient within 48 h after admission. Medication analysis of inpatient medications and drug therapy counseling were quality improvement measures, so explicit patient consent to participate in the study was not required.

### 2.5. Endpoints

#### 2.5.1. Primary Endpoint

DRP during the inpatient stay includes pharmaceutical interventions and their acceptance.

#### 2.5.2. Secondary Endpoint

Drug interactions on the second day of the inpatient stay and in the discharge medication.

#### 2.5.3. Measurement of the Primary Endpoint

DRP, type of interventions (written and oral), and results of these interventions were coded patient-specifically using the Pharmaceutical Care Network Europe (PCNE) Questionnaire on Drug-Related Problems V 8.0 [15]. The PCNE defines a DRP as an event or circumstance involving drug therapy that actually or potentially interferes with desired health outcomes [6]. The coded data were documented in a project-specific Access database and analyzed descriptively.

At the time of the study, the PCNE classification was not available in German. For this project, the PCNE classification was translated into German by two experienced pharmacists from different hospitals to ensure consistent coding. Examples were included when the translation did not clearly describe the situation. The consistency of the coding by different people was verified using patient cases.

The DRPs were coded according to the flow chart in Figure 2.

First, the DRP was classified as “potential” or “manifest”. Potential problems could “potentially” occur but have not yet affected the patient, whereas a manifest problem has already reached the patient. The problem was raised with the attending physician, and it was stressed that the medication should be adjusted in a timely manner, e.g., in the next ward round.

In the next step, each DRP was categorized into a problem type. Three different problem type classes (P1–P3) allowed a rough classification, with another seven subclasses (P1.1–P3.3) for a more detailed description. Similarly, the causes were classified into eight different categories (C1–C8) and a further 36 subcategories (C1.1–C8.3). For each DRP, up to three causes could be selected according to the specifications of the PCNE. The intervention type could then be classified into five main classes (I0–I4) and 17 subclasses (I0–I4.2). Each DRP could trigger up to three interventions at different levels. Finally, the acceptance (three main classes A1–A3, ten subclasses A1.1–A3.2) and the outcome of the intervention (four main classes O0–O3, seven subclasses O0.1–O3.4) were assessed.

The final coding of the DRP was double-checked by two pharmacists.

In the control phase, only DRPs were coded, but no intervention was allowed by the pharmacist. In the intervention phase, the DRP, the interventions as described above, and the results of the interventions were coded. Deviations from the presented optimization strategies were also counted as DRPs in the intervention phase (Figure 2).

In the official version of PCNE V8.0, which was used in this study, there was no problem type that addressed the aspect of “unclear documentation”. Thus, we added another category for this area within the problem types (P3.4) and the corresponding causes (C9).

### 2.6. Recruitment

All consecutive patients admitted to the palliative care unit during the 12-month period from 01/18 to 06/18 and 09/18 to 03/19 and who fulfilled the inclusion criteria were informed about the study during the ward round.

### 2.7. Data Collection

During the two phases, a pharmacist routinely recorded the medication of all patients admitted to the palliative care unit. Time points for medication recording were before admission, at admission, and at discharge. Medication before admission was documented based on the most recent medical record or the patient’s chart from the previous ward. Medication at admission was defined as the medication on the first full day in the palliative care unit.

In addition, DRP before admission, at admission, daily, and at discharge were documented and classified.

The data on medication and DRP were recorded in an Access database. In addition, socio-demographic data such as age, gender, and primary and secondary diagnoses were collected.

#### 2.7.1. Control Phase

During the control phase, lasting six months, the pharmacist was only an observer and was not allowed to intervene. The pharmacist regularly attended the team meetings and ward rounds to follow the progress of the patients’ illnesses. The processes in the unit were documented regarding the provision of medications. This included preparation of the medication, its use and administration, and assessment of the prescribed drugs, e.g., dosage, indication, interactions, potential side effects, economic aspects, and feasibility in the domestic environment.

#### 2.7.2. Inter-Phase

Between the two phases, an interface phase took place in which the data from the control phase was analyzed. In addition to DRP, weaknesses in the medication process were identified. Interventions and strategies for optimizing the medication process were developed with a team of pharmacists, discussed with experienced palliative care physicians and palliative care nurses, and modified if necessary. DRPs and weaknesses, as well as optimization strategies, were presented to the palliative care team.

#### 2.7.3. Intervention Phase

In the intervention phase, the pharmacist was allowed to flag DRP and other drug-associated risk factors and to make recommendations regarding drug therapy. To assess the timely extent of the pharmacist’s advice, the duration of the respective intervention was recorded. As in the control phase, the processes on the unit were documented regarding the provision of medications. The intervention phase covered a period of six months.

### 2.8. Data Analysis

The data on the primary and secondary endpoints were analyzed using descriptive statistics. Depending on the distribution of the data, a *t*-test and a Mann-Whitney U test, respectively, were used to test for differences between the two phases. *p*-values < 0.05 were considered significant. The statistical analysis was conducted with the statistical software IBM^®^ SPSS^®^ Statistics Version 26 for Microsoft^®^ Windows 10 (IBM Corp., Armonk, NY, USA, 2019). The histograms were created in Microsoft Excel.

## 3. Results

### 3.1. Study Population and Patient Flow

Overall, 309 patients were admitted to the palliative care unit in the recruitment phase: 138/148 patients were included in the control phase (January–June 2018) and 146/161 patients in the intervention phase (September 2018–March 2019) (Figure 3) The reasons for dropout in both phases are listed in Figure 3.

### 3.2. Demographic Data

Overall, patients in both phases were similar regarding gender, age, type of disease, duration of inpatient stay, and death (Table 1).

### 3.3. Drug-Related Problems

Due to the large number of coding aspects of a DRP in the PCNE classification, only those with a significant difference between the control and intervention phases are listed in this manuscript for better readability. In the control phase, 634 DRPs were coded with 571 potential and 63 manifest problems. In the intervention phase, 445 DRPs were coded, of which 383 were potential and 62 were manifest. The difference in the total number of DRPs and the number of potential problems between the phases was statistically significant (Table 2).

The most frequent manifest problems were classified as tolerable both in the control phase (50/63 DRPs; 79%) and in the intervention phase (36/62; 58%). These were predominantly insufficient or missing laxatives during opioid therapy. In the control phase, 13/63 (21%) of the manifest problems were not tolerable; in the intervention phase, 26/62 (42%). In both phases, this was mainly related to strong side effects due to an overdose of, for example, hydromorphone or antihypertensives.

### 3.4. Problems

When classifying the type of problem, a significant difference between the control and intervention phases was only in category P3 “Others” and in subclass P3.4 “Unclear Documentation” (*p* = 0.001) (Table 3).

These included

lack of indication when prescribing a drug;Infusion plans without instructions for routes of administration or assigning active substances to indications;lack of documentation of the effect of as-needed doses;use of drugs without a prior prescription.

In the control phase, each patient was affected by an average of 2.7 documentation problems, whereas in the intervention phase, there were 0.9 documentation problems per patient.

Another common DRP in both phases was “Unnecessary drug therapy” (P3.2). This DRP mainly involved the use of proton pump inhibitors. Other drugs affected were, for example, regular antiemetic medication without nausea as a symptom to be treated or the use of antihypertensives in patients with low blood pressure.

### 3.5. Causes

There was a significant difference between the phases in categories C5 “Dispensing” (*p* = 0.001) and C9 “Documentation” (*p* = 0.001) (Table 3).

The main cause in the control phase was “Documentation” (C9) (312/634, 49%), while in the intervention phase, “Dose selection” (C3) (144/634, 23%) was a major cause. Regarding the latter, in both phases, the most frequent reason was a dose that was too high (C3.2), and more than half of the corresponding DRPs involved proton pump inhibitors (control phase 51/103 DRPs, 16%; intervention phase 53/117 DRPs, 26%).

In the control phase, the main cause of a DRP was unclear documentation, with a significant difference between the two phases. The cause “Documentation” (C9) was divided into four subclasses to further describe these causes. In all of these four subclasses, there was a significant difference between the control and intervention phases (Figure 4).

The second most frequent cause of a DRP in the control phase was “Dispensing” (C5) and, in this context, the lack of required information (C5.2) (Table 3).

In almost all cases, this concerned the infusion plan (146/150 DRPs), as no route of administration was specified or no indication was provided for the syringe driver.

After analyzing the data from the control phase, the following suggestions for optimizing the medication process were developed and presented to the team: specification of the indication on admission/initiation of therapy; standardized specification of the active substance (trade name); prescription of laxative measures as “as needed” or “regular” medication; explicit documentation of the reason why no laxatives were given; consistent documentation of the effect of “as needed” medication in order to justify further administration; prioritization of “as needed” medication for one indication.

### 3.6. Planned Intervention

In the intervention phase, 182/445 DRPs (40%) were not intervened (I0) (Table 3). This related mainly to documentation (131/182 DRPs; 72%) and missing information regarding as-needed administration (perfusor or on-demand medication). These documentation issues were not addressed individually but collectively in the team meetings. Other DRPs (23/182 DRPs; 13%) for which no intervention took place in the intervention phase were related to unnecessary drug therapy (proton pump inhibitors, statins, etc.). For these DRPs, in most cases the pharmacist was not present on the ward or the discharge letter could not be reviewed by the pharmacist before discharge (12/23 DRPs; 52%).

Most frequently, interventions happened at the prescriber level (I1) in short conversations (148/256 interventions; 58%) lasting a median of five minutes (range: 5–25 min). This type of intervention mainly concerned DRPs that occurred more frequently or did not require much explanation, e.g., monitoring of side effects due to renal function impairment in, e.g., hydromorphone/metoclopramide use (43/148 interventions; 29%), proton pump inhibitor use/dosing (32/148 interventions; 22%), inadequate dosing interval (19/148 interventions; 13%), and constipation prophylaxis (8/148 interventions; 5%).

The second most frequent intervention was detailed discussions with prescribers during the ward round or after the morning hand-over (52/256 interventions at prescriber level; 20%). An intervention of this type lasts a median of 15 min (range: 5–30 min).

The third most common intervention was a written note in the patient chart (48/256 interventions at prescriber level; 19%), which was color-coded. Prescribers were asked to sign this note to indicate that they had read it and, if they did not implement the intervention, to leave a short note explaining why. The written intervention lasted a median of 10 min (range: 5–45 min).

The least used type of intervention at the prescriber level was the active request for information by the prescriber or the nurse (8/256 interventions, 3%). Most of these interventions occurred in the course of the ward round or when looking through the files. Active inquiries were mainly made by the nursing staff.

### 3.7. Acceptance of Interventions

In the intervention phase, 227/256 (87%) interventions were accepted by the prescriber, the nursing staff, or the patient (A1). Approximately 26/256 (10%) of the interventions were not accepted (A2). Most frequently, this was related to proton pump inhibitor therapy, for which no indication was given or the dosage was incorrect. In 182/192 (43%) cases, no intervention took place, which is why the acceptance of this intervention remained open (A3) (Table 3).

### 3.8. Status of the Problem

There was a significant difference between the control phase and the intervention phase with regard to unsolved problems (*p* = 0,001). The pharmacist’s intervention resulted in significantly fewer problems not being solved. Even without this pharmacist intervention, 146/634 DRPs (23%) were completely solved (O1.1) and 28/634 DRPs (4%) were partially solved (O2.1) in the control phase. In the intervention phase, 196/445 DRPs (44%) were completely solved (O1.1) and 30/445 DRPs (7%) were partially solved (O2.1) (Table 3).

Failure to resolve a problem due to prescriber refusal was only possible in the intervention phase. This happened in 40/218 unresolved DRPs. Most notably, patients were prescribed the medication prior to their inpatient stay in the palliative care unit. Prescribers in the study were often unsure of the intervention by the pharmacist because the prescription was just taken over. In 7 out of 217 cases (3%) the intervention was not effective and the DRP could therefore not be resolved, for example, unrelieved delirium before the patient died or insufficient symptom control of dyspnea.

## 4. Discussion

Drug therapy always involves certain risks for patients and prescribers, especially in palliative care when patients are very sick and time is limited. Our study demonstrates that drug-related problems in a palliative care unit can be identified, addressed, and reduced through the systematic and continuous monitoring of the medication process by a pharmacist. This can potentially lead to an increase in the effectiveness and safety of drug therapy.

The involvement of pharmacists in ‘medication review and reconciliation’, ‘medication counselling, education, and training’, ‘administrative roles’, ‘direct patient care’, and ‘education and scholarship’ in palliative care has been described [16]. Although the value of pharmacists in these roles has been demonstrated in other settings, this evidence is still largely lacking in palliative care. In Germany, for example, the presence of clinical pharmacists is not yet the norm [17]. The added value of a pharmacist, e.g., through a before-and-after study like ours, is therefore important in discussions with administrative structures and funding bodies. The direct comparison of our results with other palliative care trials is difficult, partly because of the heterogeneity of the endpoints and tools used.

In the study presented here, an average of four DRPs per patient were recorded in the control phase, where, according to the study, no intervention by the pharmacist was allowed. The number of DRPs in the control phase was probably even higher, as some information on drug therapy was missing (2,7 DRP per patient regarding unclear documentation). For example, in the control phase, indications were not routinely recorded in the patient’s medical record, limiting the possibility of assessing dosages. The importance of documentation of the indication was therefore emphasized to the team in the interphase.

In the intervention phase, the number of DRPs was significantly reduced to an average of three per patient for whom interventions were recommended to resolve the DRP (0.9 DRP per patient with unclear documentation). In the previous study by Rémi et al., an average of five DRPs per patient could be identified for which the pharmacist intervened [6]. The collection of numerous DRPs with respective interventions, as in the present study, confirms the findings of earlier international studies [18,19]. In these, the positive influence of pharmacists in palliative care in the recording and resolution of DRPs and the associated improvement in the quality of inpatient care were demonstrated. Involving a pharmacist in the home care setting in a study by Hussainy et al. also led to the identification of numerous DRPs in medication analyses (120 DRP in 46 analyses) [20].

Although the difference between the two phases of the study was not significant, the high number of unnecessary drug treatments identified as a problem again highlights the role that pharmacists can play in helping to assess which drugs should be discontinued. The need for deprescribing to reduce burdensome medication has been addressed in several studies [21,22]. As already mentioned, the lack of documentation of the indication is the main cause of DRP, especially in the control phase. We believe that it is essential for evaluating medications, not only in the palliative care context. Indications are assumed to be logically derived from the prescribed medication. However, at the latest, when several people are involved in patient care, this derived indication is subject to individual interpretation. Equally important to the evaluation of drugs—especially in a specialty such as palliative care, where the focus is on the treatment of distressing symptoms—is the assessment and subsequent documentation of an effect. Both aspects could be significantly improved by the pharmacist.

The major difference between the previous results and our study is the before-and-after comparison, which allows the impact of the pharmacist to be assessed directly.

Due to the constant presence of the pharmacist on the ward, DRP could be identified promptly, and appropriate interventions could take place, thus quickly resolving the DRP and avoiding negative consequences for the patient. The high acceptance of the interventions in the study (87%) can be interpreted as acceptance by the pharmacist as part of the palliative care team. The positive impact of pharmacists on medication, as well as their high level of acceptance within the team, has been observed by others [23,24].

In the present study, there were many different types of communication between the pharmacist and the palliative care team. The pharmacist identified the DRP and decided on the type of intervention depending on the potential risk. Potential problems, especially documentation problems or dosing intervals, were discussed directly with the prescribers. If the information about the DRP or the corresponding intervention was relevant for the physician and the nursing staff, this was documented in writing in the patient record to reach as many members of the staff as possible with this intervention. Manifest DRPs requiring action (e.g., dosages far outside the approved dose range, severe side effects, etc.) were actively discussed with the physician during ward rounds. A combination of face-to-face conversations and written information achieved the highest acceptance, and we would recommend it as the best solution for the daily routine on the ward. In the direct discussion, the clinical relevance of the intervention could be clearly emphasized and potential questions clarified. The acceptance or non-acceptance of the intervention was immediately recognized, and in the case of non-acceptance, this could be directly justified, and the pharmacist had the opportunity to suggest a potential alternative.

The main task of the pharmacist in our study was the detection, resolution, and prevention of DRPs. Based on this, advice for the physicians and, in particular, support in drug selection and dosage were essential. These tasks were comparable to those in other German studies [6,23]. In the present study as well as in other international studies, interventions by the pharmacist set their focus mainly on the identification of untreated symptoms and assessing the appropriate indications for the medications administered [16].

Hussainy et al. demonstrated in 2011 that pharmacists can primarily take this counseling position in a palliative care team, which was also confirmed in the study presented here [20]. Pharmacists’ knowledge of medications helps to minimize medication errors, potentially improves symptom management, and raises the awareness of the team about the use of drugs in general. The pharmacist supports the palliative care team and the patients by sharing his or her expertise on the respective drug therapies. This recent review by Wernli et al. demonstrated that pharmacist-led interventions can further decrease the length of stay, improve patients symptoms, and save doctors and other staff members time on pharmacy issues [16].

Nevertheless, a clear division of tasks is recommended in order to facilitate and support the interdisciplinary palliative care team in daily life and not only in acute needs.

The role and responsibilities of clinical pharmacists can vary by region and healthcare system, so it’s essential to consider the specific context in which they work. In Germany, the role of the clinical pharmacist is not clearly defined and varies from setting to setting and institution to institution. The pharmacist can only make recommendations and does not have the authority to prescribe. The physician is responsible for prescribing treatment. When it comes to the use of pharmacists on hospital wards, Germany is lagging behind other European countries [17]. Only recently has the role of ward pharmacist been enshrined in law in some parts of Germany. The staffing and associated costs are offset by savings from the avoidance of adverse drug reactions, the optimization of therapy, and the rational use of medicines. The nationwide use of ward pharmacists in Germany is hampered by the fact that the DRG system does not provide for remuneration of the pharmacist’s services. Nevertheless, there are a number of hospitals where pharmacists are directly involved in the medication process [17]. Slowly, however, changes are happening. Since 2022, certain pharmaceutical services, such as advice on polymedication or oral tumor therapies, have been subject to special remuneration for retail (but not hospital) pharmacies [25].

Addressing drug-related problems often involves collaboration among healthcare providers, including physicians, pharmacists, nurses, and other members of the healthcare team. Pharmacists, in particular, play a critical role in identifying, preventing, and resolving drug-related problems through medication therapy management and patient counseling.

### Limitations of the Study

The aim of this study was to directly compare the impact of the pharmacist on the number of DRPs. To ensure the feasibility of the study with this patient population, a before-and-after study was chosen. The palliative patient population in this study was typically heterogeneous in age, gender, palliative stage, and underlying disease. In two consecutive study phases, this heterogeneous patient population may affect the comparability of the data. A randomized study design would be necessary to achieve this comparability. However, this would require a much larger patient population, which was not feasible due to limited resources.

The approach of two consecutive phases in which the pharmacist first came to the ward to observe (before the situation) and then moved to the intervention phase (after the situation) was generally feasible. However, it carries a certain risk of bias, as it can be assumed that the mere presence of the pharmacist on the ward changed the behavior of the interdisciplinary team. The doctors confirmed that drug therapy was discussed more frequently since the pharmacist was present on the ward. We could have analyzed the prescribing data retrospectively, but since we wanted to assess the whole medication process, we decided to have the pharmacist participate as a silent observer.

In order to obtain a larger patient population and to account for institutional differences, a multicenter trial is recommended for a follow-up study.

## 5. Conclusions

Drugs are part of everyday symptom management in palliative medicine. Treatment approaches in palliative care are oftentimes highly individualized, focusing on symptom management and improving the quality of life of patients with advanced illnesses. Drug-related problems are therefore hardly avoidable. For the patients’ safety, efforts should be made to identify and minimize DRPs at an early stage. Pharmacists can contribute significantly to this. It is not easy to assess the overall impact of the interventions presented here on symptom control. Palliative care is an interdisciplinary field with a heterogeneous patient population. Admission to a palliative care unit and the multiprofessional care provided there is a complex intervention, and an improvement in symptom control or an expected overall deterioration of the patient in the context of the underlying disease are normal outcomes in our patient population. The exact contribution of pharmaceutical interventions to the overall effect is therefore difficult to measure. However, our study provides another piece of the puzzle in assessing the contribution of clinical pharmacists and represents at least one increase in patient safety.

## Figures and Tables

**Figure 1 pharmacy-11-00160-f001:**
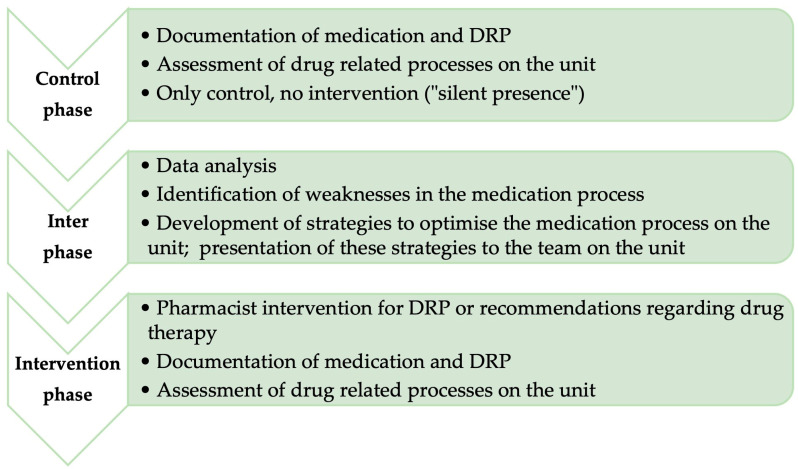
Flowchart study schedule.

**Figure 2 pharmacy-11-00160-f002:**
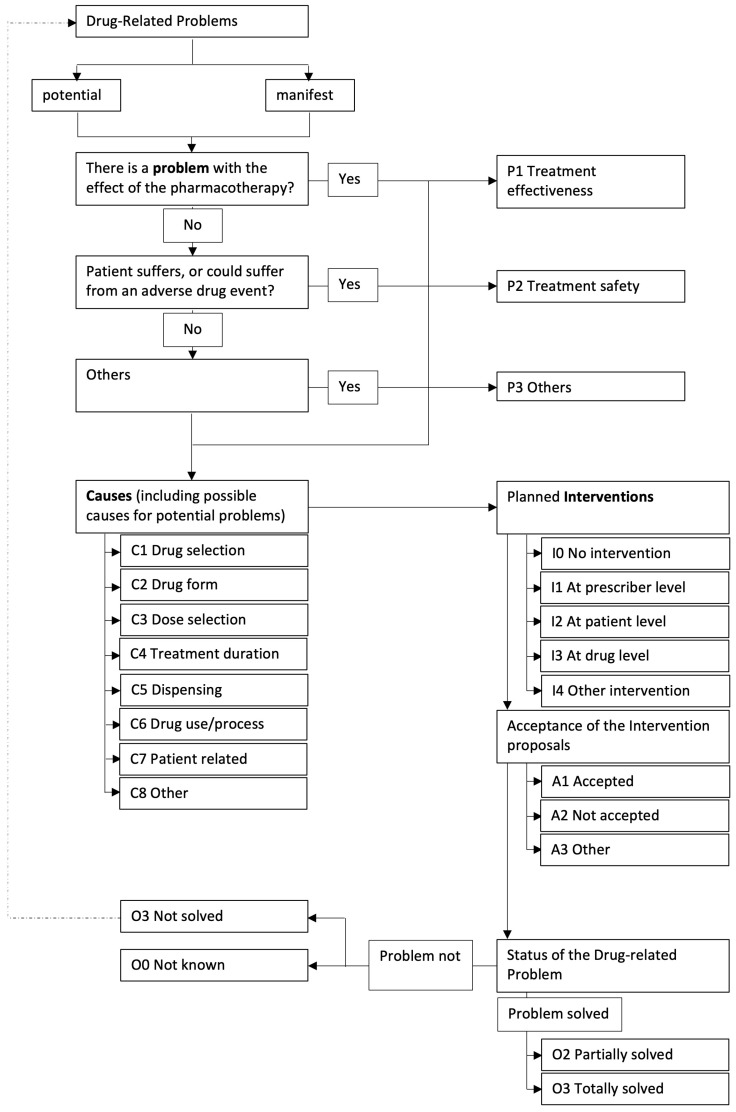
Flowchart for the coding of DRP.

**Figure 3 pharmacy-11-00160-f003:**
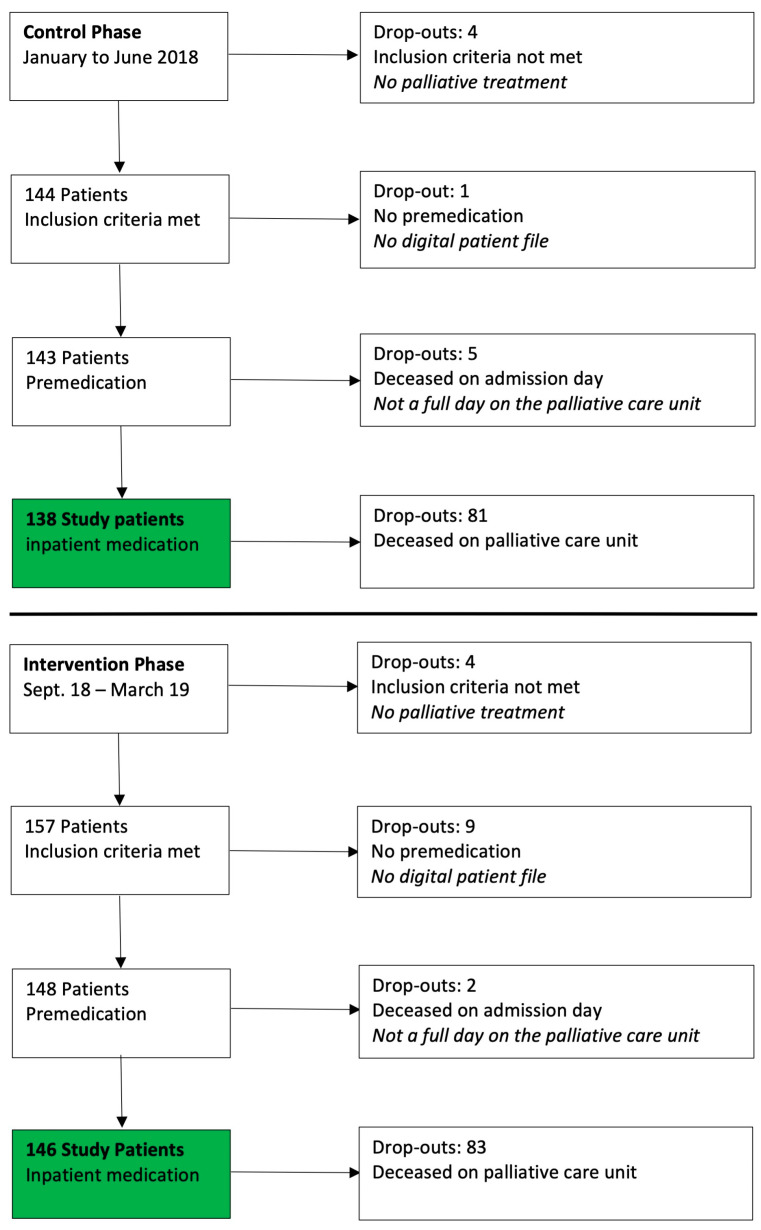
Patient flow control and intervention phase.

**Figure 4 pharmacy-11-00160-f004:**
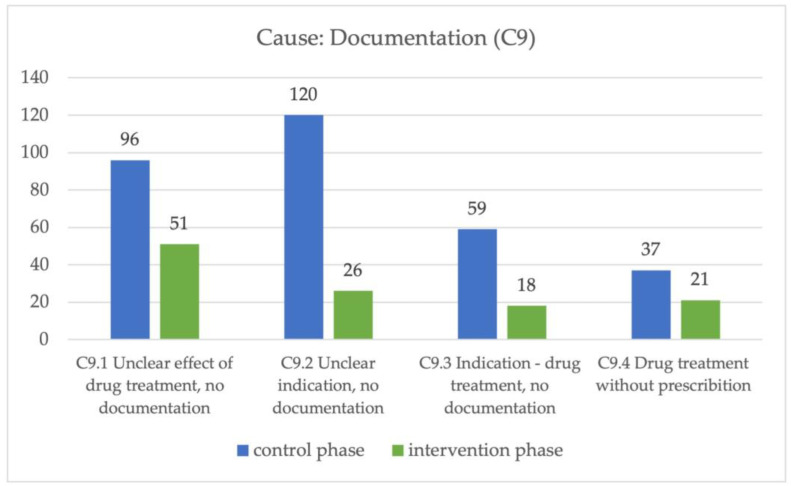
Cause “documentation” in control and intervention phases.

**Table 1 pharmacy-11-00160-t001:** Demographic data of the patients in the study.

	Control Phase n = 138	Intervention Phase n = 146	*p*-Value
Female [n] (%)	64 (46)	80 (55)	0.157
Age [years mean] (standard deviation)	72.5 (11.4)	70.3 (12.5)	0.156
>65 years [n] (%)	111 (80)	95 (66)	0.449
Malignant disease	121 (87%)	128 (87%)	1.000
Non-malignant disease	15 (10%)	17 (12%)	1.000
Inpatient stay			
Duration of inpatient stay [days Median] (range) incl. Day of admission	10 (1–52)	9.5 (1–35)	0.934
Died during inpatient stay [n] (%)	81 (59)	83 (57)	1.000

**Table 2 pharmacy-11-00160-t002:** Number of DRPs in both phases, * significant difference.

DRP [Median] (Range)	Control Phase	Intervention Phase	*p*-Value
total	634 [4] (0–14)	445 [3] (0–10)	*0.005 **
potential	571 [4] (0–13)	383 [3] (0–8)	*0.001 **
manifest	63 [0] (0–4)	62 [0] (0–3)	0.555

**Table 3 pharmacy-11-00160-t003:** Results, * significant difference.

	Control Phase	Intervention Phase	*p*-Value
**DRP**	634	445	** *0.001 ** **
**Problems**			
P3 Others [n] (%)	463 (73)	253 (57)	** *0.001 ** **
P3.2 Unnecessary drug treatment [n] (%)	72 (11)	89 (20)	not significant
P3.4 Unclear documentation [n] (%)	367 (58)	136 (31)	** *0.001 ** **
**Causes**			
C3 Dose selection [n] (%)	144 (23)	180 (40)	not significant
C3.2 Drug dose too high [n] (%)	103 (16)	117 (26)	not significant
C5 Dispensing [n] (%)	155 (24)	55 (12)	** *0.001 ** **
C5.2 Necessary information not provided [n] (%)	150 (24)	47 (11)	** *0.001 ** **
C9 Documentation [n] (%)	312 (49)	116 (26)	** *0.001 ** **
C9.1 Unclear effect of drug treatment, no documentation [n] (%)	96 (15)	51 (11)	** *0.001 ** **
C9.2 Unclear indication, no documentation [n] (%)	120 (19)	26 (6)	** *0.001 ** **
C9.3 Indication—drug treatment, no documentation [n] (%)	59 (9)	18 (4)	** *0.001 ** **
C9.4 Drug treatment without prescription [n] (%)	37 (6)	21 (5)	** *0.004 ** **
**Planned Interventions**			
I0 No intervention [n] (%)	634 (100)	182 (41)	not significant
I1 At prescriber level [n] (%)	0 (0)	256 (58)	not significant
**Acceptance of the Intervention proposals**			
A1 Intervention accepted (by prescriber or patient) [n] (%)	0 (0)	227 (87)	not significant
A2 Intervention not accepted (by prescriber or patient) [n] (%)	0 (0)	26 (10)	not significant
A3 Other (no information on acceptance) [n] (%)	634 (100)	192 (43)	not significant
**Status of the DRP**			
O0.1 Problem status unknown [n] (%)	9 (1)	2 (0)	not significant
O1.1 Problem totally solved [n] (%)	146 (23)	196 (44)	not significant
O2.1 Problem partially solved [n] (%)	28 (4)	30 (7)	not significant
O3 Problem not solved [n] (%)	453 (71)	0 (0)	** *0.001 ** **

## Data Availability

This article contains data from the doctoral thesis of L.K. published at the Faculty of Medicine, Ludwig Maximilian University, Munich, Germany.

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
