# Peer review of "Drug Therapy Safety in Palliative Care—Pharmaceutical Analysis of Medication Processes in Palliative Care"

_pharmacy, 2023, doi:10.3390/pharmacy11050160_

Round 1

Reviewer 1 Report

The authors have developed and ran a study on drug-related problems in a clinical setting. The authors have followed the STROBE checklist in order to methodically approach data reporting and interpretation. The manuscript is well-written and coherent. There are, however, several issues that need to be resolved before this manuscript is considered for publication.

1. There are some terrible formatting errors in the submitted manuscript - tables are repeated numerous times (as well as Figure 4), some figures are truncated, captions are missing etc. This makes the manuscript difficult, if not impossible, to thoroughly evaluate in terms of accuracy and relevance of the reported results.

2. Apparently there is a reference missing on line 178.

3. Some of the results are missing from the tables (i.e. table 3) and this makes following the data really difficult.

4. The paper is significantly under-referenced and the Discussions chapter needs to be further developed. The authors do not address all the results of the study in comparison with literature data and this can be confusing when trying to understand the relevance of the study (population selection, addressability, magnitude of intervention).

5. The paper requires a paragraph in the last part of Discussions where limitations of the study should be addressed.

6. Did the authors consider whether the two lots (control and intervention) could be different in respect to other parameters than age, sex and category of disease? 

7. The conclusions are a bit forced compared to the magnitude of the results and the quality of the discussions and should be rephrased accordingly. The authors have not performed a multi-centric nor a public healthcare study that would be able to properly assess the role of the pharmacist in the holistic care of patients.

8. The histograms in Figure 4 are made in Microsoft Excel, however, this is not mentioned in subsection 2.8. If SPSS was used for the analysis as claimed, why did the authors not include the associated graphs in SPSS?

Respectfully submitted,

Overall, I have not identified grammar errors, but some phrasing could be improved. For instance "Other drugs affected were" or ""Potential problems could "potentially" occur.

Author Response

The authors would like to thank the reviewers for the critical examination of the manuscript, contributing to its further improvement. Our study aimed to identify DRP in a palliative care unit (PCU) and to evaluate corresponding pharmaceutical interventions. According to the suggestions, we have thoroughly revised our manuscript and its final version is enclosed as well as our comments to the reviewers.

Reviewer 2 Report

Drug Therapy Safety in Palliative Care Pharmaceutical Analysis of Medication Processes in Palliative Care

The authors have written about an understudied and very important topic. This topic is highly relevant to this journal. Considering the lack of objective studies and the recent controversies on this issue in some countries against pharmacists as ward (ambulatory) team members in treatment, the paper provides important information from evidence-based medicine that could be considered a bridge between real practice (treatment) and guidelines (recommendations). The purpose of the research is very well-defined, and I'm sure the objectives will be met. Generally, the paper has medium issues with the writing standard and tables are without manipulation.

Despite these positive issues, the paper has many important limitations that should be discussed (e.g. small sample size, selection criteria, poor comparison with other papers, poor introduction and discussion), and methodological flows and interventions should be better described.

However, in the current form presented, it requires a deep major revision before consideration for publication.

The manuscript could be strengthened by attending to the following matters:

GENERAL COMMENTS:

Positive:

-        Important topic

-        Novelties

-        Clinical topic

Negative:

-        Poor methodology

-        Many typo errors

-        Some of the parts should be modified completely

SPECIFIC COMMENTS:

Abstract:

Also = delete

before-after study in a PCU starting with a control phase = unclear methodology (e.g., cohort, case-control study?)

Professionals = prescriber? Physicians?

Summary = Please modify according to the results

Keywords: Recheck them (e.g. MESH keywords are recommended).

1.      Introduction

This part of the manuscript should be expanded. The authors described some studies but did not mention that clinical pharmacists within the daily rounding (e.g., rounds) could solve more problems compared to the control group (Positive evidence for clinical pharmacist interventions during interdisciplinary rounding at a psychiatric hospital. Sci Rep. 2021 Jul 1;11(1):13641. doi: 10.1038/s41598-021-92909-2). This issue should also be researched by the palliative care team.

Specifics in the palliative treatment should be described here (e.g., irrational polypharmacy).

2. Materials and Methods

2.1 Design

Unclear why the authors included the interphase …

Unclear type of the study (e.g., pre-post versus cohort versus case-control).

An appropriate STROBE checklist should be used, including an appropriate study design.

The authors should describe the exact type of clinical pharmacists’ interventions.

2.2 Setting

The authors should describe how clinical pharmacists collaborate in daily activities and the responsibilities and roles in the context of German legislation (e.g., the attending physician is responsible for the whole treatment).

The authors should describe reimbursed clinical pharmacy services in Germany.

The multi-professional team should include a clinical pharmacist.

2.3 Population

Fluent in English? Did the authors validate PCNE Q in German language?

2.5 Endpoints

The definition of DRP is missing.

The endpoint should be included in the abstract.

Many typo errors = statistically significant (

Figures are not adapted to the text. Text is confused and mixed with pictures.

2.6 Recruitment

Effect size? Did the authors calculate the proposed effect size?

How did they decide the exact number of participants?

2.7 Data collection

Is medication reconciliation provided in Germany? It has an impact on the final results.

2.7.1 Control phase

It is not clear how the selected patients are and the exact number.

2.7.2 Interphase

It is unclear why the authors included this phase (e.g., not included in STROBE).

2.7.3 Intervention phase

The authors should better describe the interventions and clinical pharmacist as a part of the multi-disciplinary team on the ward.

RESULTS

Too many tables … The authors should modify the results and include only the most important results in a picture form (e.g., move tables to the Appendix).

Many typo errors = (Fehler! Verweisquelle konnte nicht

DISCUSSION

This part of the manuscript should be expanded.

1.      Main results and comparison

2.      Improvements

3.      Reimbursement

4.      Limitations

How is clinical pharmacy regulated in Germany? Please check the following reference: Recommendations for broader adoption of clinical pharmacy in Central and Eastern Europe to optimise pharmacotherapy and improve patient outcomes. Front Pharmacol. 2023 Aug 2;14:1244151. Doi: 10.3389/fphar.2023.1244151. The authors discussed important points regarding the reimbursement of clinical pharmacy services.

Author Response

(The authors gave the same response as above.)

Round 2

Reviewer 1 Report

The authors have significantly improved the quality of the manuscript. Still, some grammar/spelling errors remain, including in the newly inserted text. However, the paper is now acceptable and up to standards.

Some grammar/spelling errors remain, including in the newly inserted text.

Author Response

The authors would like to thank the reviewers for the re-examination of the manuscript. According to the suggestions, we have revised our manuscript and its final version is enclosed as well as our comments to the reviewers.

Reviewer 2 Report

The authors accepted almost all of my proposed recommendations and significantly improved their paper.

In addition there are some shortcomings which should be solved before acceptance:

Discussion:

Lines 368-377: In the intervention phase, the number of DRPs was significantly reduced...

The authors could compare these results with the other pharmaceutical services (e.g., in Slovenian in the reimbursed medication review clinical pharmacists provided app 5 recommendations per patient and acceptance rate is 50%-outpatients). Clinical pharmacist interventions in cardiovascular disease pharmacotherapy in elderly patients on excessive polypharmacy : A retrospective pre-post observational multicentric study. Wien Klin Wochenschr. 2021 Aug;133(15-16):770-779. 

Lines 431-435 Other studies .... Authors included only one reference (N#16), please add more if you used "other studies" next to the reference N#16

Lines 435-451: >The role and responsibilities of clinical pharmacists can vary... The authors should include some relevant papers, including a "state-the-art" paper about clinical pharmacy in the Central Europe: Recommendations for wider adoption of clinical pharmacy in Central and Eastern Europe in order to optimise pharmacotherapy and improve patient outcomes. Front Pharmacol. 2023 Aug 2;14:1244151. doi: 10.3389/fphar.2023.1244151. 

It would be nice for the readers if the authors comment on this in the context of German development and roles of clinical pharmacists. 

Author Response

(The authors gave the same response as above.)
